# $\delta^2$-EXPLORATION FOR REINFORCEMENT LEARNING

## ABSTRACT

Effectively tackling the *exploration-exploitation dilemma* is still a major challenge in reinforcement learning. Uncertainty-based exploration strategies developed in the bandit setting could theoretically offer a principled way to trade off exploration and exploitation, but applying them to the general reinforcement learning setting is impractical due to their requirement to represent posterior distributions over models, which is computationally intractable in generic sequential decision tasks. Recently, *Sample Average Uncertainty (SAU)* was develop as an alternative method to tackle exploration in bandit problems in a scalable way. What makes SAU particularly efficient is that it only depends on the value predictions, meaning that it does not need to rely on maintaining model posterior distributions. In this work we propose $\delta^2$-*exploration*, an exploration strategy that extends SAU from bandits to the general sequential Reinforcement Learning scenario. We empirically study $\delta^2$-exploration in the tabular as well as in the Deep Q-learning case, proving its strong practical advantage and wide adaptability to complex reward models such as those deployed in modern Reinforcement Learning.

## 1 INTRODUCTION

In Reinforcement Learning (RL) an agent interacts with an external environment taking sequences of actions that cause transitions between states of the environment, with the scope to maximize the sum of rewards that are gathered at each state. In a typical scenario, the RL agent does not initially posses perfect knowledge of the consequences of its actions on the environment, but has to learn that through experience. It therefore has to face the classical *exploration-exploitation dilemma*, i.e. deciding whether to exploit the actions that are known to maximize immediate reward or whether to explore unfamiliar actions and states in order to potentially find ways to increase future rewards.

Efficient exploration requires the agent to quantify the accuracy of its current estimates of the state-action values (the conditional expected rewards), so as to trade off expected rewards and uncertainty in a principled way (Strehl & Littman, 2008; O'Donoghue et al., 2018). Measuring the uncertainty associated with the value of each state-action is thus a key component of conventional algorithms for addressing the exploration-exploitation dilemma. These ideas have been developed into successful uncertainty-based exploration strategies such as the *Upper Confidence Bound (UCB) algorithm*, which quantifies uncertainty through confidence intervals (Auer et al., 2002), and *Thompson Sampling* (TS), which instead models the posterior distribution over values (Thompson, 1933). However, obtaining either of these measures of uncertainty for general sequential Reinforcement Learning problems has proven arduous because of their reliance on a measure of uncertainty around the current value predictions, which is typically infeasible to obtain for complex environments and value functions. This difficulty limits the applicability of explorations strategies based on UCB and TS for general RL problems, compromising in particular their adaptability to algorithms such as Q-learning (Watkins, 1989; Watkins & Dayan, 1992) and DQN (Mnih et al., 2015), which consequently typically resort to addressing exploration via inefficient heuristics like $\epsilon$-greedy action selection.

Recently, Rigotti & Zhu (2021) tackled the mentioned limitations of conventional exploration strategies in the constrained but interesting case of bandit problems. Central to that effort was the development of a novel uncertainty measure, the *Sample Average Uncertainty (SAU)*, which departs from the approach of estimating the uncertainty of expected rewards of selected actions. In contrast to what is done in UCB and TS, SAU is an uncertainty measure that only depends on the value prediction of each action. Specifically, it consists in quantifying the variance of sample averages of rewards under the assumption of known values, which can be used to simply construct an estimated

variance by plugging in the estimated values. Finally, Rigotti & Zhu (2021) were able to use SAU to developed two SAU-based exploration strategies, *SAU-UCB* and *SAU-Sampling*, which were shown to display state-of-the-art performance in multi-armed bandits, contextual bandit, and deep bandits.

In this work, we exploit the simplicity of SAU in Rigotti & Zhu (2021) and extend it from the bandits to the more general sequential RL setting. This extension results in $\delta^2$-*exploration*, a class of exploration strategies for RL which share the implementation simplicity of SAU for bandits as well as its effectiveness. Our $\delta^2$-*exploration* algorithms are implementable with a minimal computational overhead that is comparable to $\epsilon$-greedy exploration, while being empirically more effective.

As a practical demonstration, we apply $\delta^2$-*exploration* to Q-leaning (Watkins, 1989). We then show that the resulting strategies display empirically efficient exploration, which very quickly reach near optimal policies. As mentioned, in terms of computation and memory requirements, these exploration strategies do not add any additional cost compared to for instance a simple $\epsilon$-greedy action selection strategy. Finally, we apply SAU-based exploration to Deep Q Networks (DQN), and observe convincing performance gains over standard DQN in several games we tested from the Atari 2600 domain.

Thanks to its efficiency, flexibility and implementation simplicity, we argue that $\delta^2$-exploration could serve as a competitive baseline and as a simple and straight-forward replacement to $\epsilon$-greedy action selection in a wide class of RL scenarios.

In the remainder of this paper, Section 2 summarizes common exploration strategies in bandits and Reinforcement Learning in order to set the stage for $\delta^2$-exploration. Section 3 establishes the notation and states the problem. Section 4.2 presents the SAU measure, and summarized SAU-based exploration strategies in the setting of multi-armed bandits as developed in Rigotti & Zhu (2021). In Section 5 we then extend these strategies to the sequential RL setting and develop $\delta^2$-exploration. Section 6 includes empirical evaluations showing substantial performance improvements over conventional exploration strategies in Q-learning and DQN. Section 7 concludes the paper.

## 2 RELATED WORK

The most common exploration strategy in Reinforcement Learning (RL) is arguably $\epsilon$-greedy action selection, which consists in following the action with the highest estimated value with high probability, but occasionally with a small fixed probability $\epsilon$ choosing any other available action uniformly at random. Despite the clear inefficiency of not differentiating lower value actions based on their uncertainty or even their value, because of its simplicity $\epsilon$-greedy action selection is still the exploration strategy of choice in RL.

One of the most popular and efficient exploration algorithms is the *Upper Confidence Bound (UCB) algorithm*, which is very successful in the bandit setting (Auer et al., 2002; Auer, 2003; Rusmevichientong & Tsitsiklis, 2010; Abbasi-Yadkori et al., 2011; Abbasi-Yadkori, 2012; Perchet & Rigollet, 2013; Slivkins, 2014). This approach is based on the principle of *optimism in the face of uncertainty*, which elegantly addresses the exploration-exploitation trade-off by maintaining confidence intervals for action-value estimates and choosing actions optimistically within these intervals. Unfortunately, UCB is difficulty to apply to RL problems on complex domains (Jaksch et al., 2010; Wen & Van Roy, 2013) due to the complexity of computing its uncertainty measure.

A related class of strategies to UCB are count-based methods (Strehl & Littman, 2008; Bellemare et al., 2016) that directly turn the counts of state-action visitations into reward bonuses to encourage the exploration of less visited ones. Despite performing near-optimally for small discrete Markov decision processes, count-based methods suffer the curse of dimensionality in high-dimensional environments (although see Tang et al. (2017) that proposed a possible high-dimensional extension using learned hashing to discretize the state space).

Another popular exploration strategy is *Thompson Sampling* (TS) which was introduced by Thompson (1933). This is a Bayesian approach that follows the principle of *sampling in the face of uncertainty*, i.e. it samples action-values from a known posterior distribution, then selects the action with the highest sampled value. TS has been successfully applied in bandit settings (Chapelle & Li, 2011; Agrawal & Goyal, 2013; Russo & Van Roy, 2014). It has also been applied to simple Reinforcement Learning problems (Strens, 2000; Osband et al., 2013). However, because calculating the posterior

distributions over values is intractable in the general case, implementations of TS in even moderately sophisticated domains require the application of posterior approximation methods, like for instance in Osband et al. (2016), which, at least in the contextual bandit setting, are known to be inefficient (see Riquelme et al. (2018)).

## 3 PROBLEM FORMULATION

We recall the definition of a Markov Decision Process (MDP) as a set of states $\mathcal{S}$ and actions $\mathcal{A}$ so that, given a current state $s \in \mathcal{S}$ and an action $a \in \mathcal{A}$, the probability of ending up in the next state $s' \in S$ is $P_{ss'}(a)$, where $P : \mathcal{S} \times \mathcal{A} \times \mathcal{S} \to [0, 1]$ is a fixed state transition distribution with $\sum_{s'} P_{ss'}(a) = 1$. Moreover, the reward $r$ is drawn from a reward distribution with $\mathbb{E}[r|s, a, s'] = R_{ss'}(a)$. The optimal value function $Q^*(s, a)$ solves an MDP by satisfying the following set of *Bellman equations* for a given discount factor $\gamma \in [0, 1)$:

$$Q^*(s, a) = \sum_{s'} P_{ss'}(a) \left[ R_{ss'}(a) + \gamma \max_{a' \in \mathcal{A}} Q^*(s', a') \right] \qquad \forall s, a.$$

MDPs can be solved using *Q-learning* (Watkins, 1989). Let $Q(s, a)$ be the current estimate of the value of action $a$ in state $s$, and let $s'$ be the state that follows $s$ upon taking $a$, and $r$ be the corresponding reward. Then *Q-learning* consists in updating the action-value function using a learning rate $\alpha$ as follows:

$$Q(s, a) \leftarrow Q(s, a) + \alpha \left[ r + \gamma \max_{a' \in \mathcal{A}} Q(s', a') - Q(s, a) \right].$$

In this update rule, Q-learning uses the $\max$ operator as an estimate the value of the next state. The term in the squared brackets

$$\delta = r + \gamma \max_{a' \in \mathcal{A}} Q(s', a') - Q(s, a) \tag{1}$$

is known as the *TD-error*, because it is a generalization of the reward prediction error in the case of a Temporal-Difference learning algorithm.

A last element that is crucial for the Q-learning algorithm to converge to the optimal value function is *exploration*. That is why Q-learning is typically implemented in combination with $\epsilon$-*greedy action selection*, a simple and rather successful exploration strategy proposed in Sutton & Barto (2018) that prescribes to fix a small probability $\epsilon$ and choose an action at time $t$ as follows:

$$a_t = \begin{cases} a \in \mathcal{A} \text{ uniformly at random,} & \text{with probability } \epsilon; \\ a^* = \arg\max_{a \in \mathcal{A}} \{Q(s_t, a)\}, & \text{with probability } 1 - \epsilon. \end{cases} \tag{2}$$

As mentioned in the previous section, $\epsilon$-greedy exploration has the obvious inefficiency that, during an exploratory event of probability $\epsilon$, actions are selected without differentiating on their uncertainty or even their value. But because of its simplicity $\epsilon$-greedy action selection is typically the exploration strategy of choice in many successful RL applications, such as Mnih et al. (2015).

## 4 SAMPLE AVERAGE UNCERTAINTY FOR CONTEXTUAL BANDITS

The Sample Average Uncertainty (SAU) is a measure of uncertainty that was recently proposed as a simple and scalable mechanism to promote exploration in the bandit setting (Rigotti & Zhu, 2021). For ease of exposition we introduce SAU from the contextual bandit perspective and notation.

### 4.1 CONTEXTUAL BANDITS

In contextual bandits at each step $t$ the agent observes a context vector $s_t \in \mathcal{X}$, selects an action $a_t$ from a set $\mathcal{A}$, after which a reward $r_t$ corresponding to the chosen action is received. The value of an action $a$ in context $s_t$ is defined as the expected reward given that $a$ is selected, and is assumed to be a fixed but unknown function of $s_t$ and $a$: $\mathbb{E}[r_t|a_t = a] = \mu(s_t, a)$. The goal is to design a decision-making policy $\pi : \mathcal{X} \to \mathcal{A}$ that maximizes the expected reward. This goal is readily quantified in terms of minimizing *expected regret*, where, if action $a_t$ is chosen at step $t$ after observing context $s_t$, we say that we incur expected regret $\max_{a' \in \mathcal{A}} \{\mu(s_t, a')\} - \mu(s_t, a_t)$, which is the difference between the optimal value received by playing the optimal action and the value received following the chosen action.

### 4.2 SAMPLE AVERAGE UNCERTAINTY EXPLORATION IN BANDITS

Let $\mathbb{T}_a$ denote the set of time steps when action $a$ was chosen so far, and let $n(a)$ be the size of this set. Assume there is a sequence of estimators $\{\hat{\mu}(s_t, a_t)\}_{t \in \mathbb{T}_a}$ of $\mu(s, a)$. The Sample Average Uncertainty (SAU (Rigotti & Zhu, 2021)) statistic $\tau(a)^2$ is defined as

$$\tau(a)^2 = \frac{1}{n(a)} \sum_{t \in \mathbb{T}_a} e_t^2 \qquad \text{with} \quad e_t = r_t - \hat{\mu}(s_t, a_t). \tag{3}$$

Obviously, computing the SAU measure only requires the prediction residuals $e_t = r_t - \hat{\mu}(s_t, a_t)$, without any need to model or access the uncertainty of $\hat{\mu}(s_t, a_t)$.

Reference Rigotti & Zhu (2021) combined this uncertainty measure with UCB and Posterior Sampling exploration and proposed two exploration strategies for contextual bandits named SAU-UCB and SAU-Sampling.

**SAU-UCB.** SAU-UCB (Rigotti & Zhu, 2021) consists in the use the SAU uncertainty measure $\tau(a)^2$ as the exploration bonus in the principle of "optimism in the face of uncertainty". This works as follow: given value predictions $\hat{\mu}(s_t, a)$ at step $t$, define $Y(s_t, a) = \hat{\mu}(s_t, a) + \sqrt{n(a)^{-1}\tau(a)^2 \log t}$, where $n(a)$ is the times action $a$ was taken. Then choose the action by maximizing over that: $a_t = \arg\max_{a \in \mathcal{A}} \{Y(s_t, a)\}$.

**SAU-Sampling.** Thompson Sampling (TS) promotes exploration by sampling value estimates from a modelled posterior distribution. The SAU-Sampling algorithm (Rigotti & Zhu, 2021) also adopts this principle of "sampling in the face of uncertainty", but unlike TS it does so by sampling values from a parametric Gaussian distribution with a mean given by the prediction and a variance proportional to $\tau(a)^2$, resulting in values $Y(s_t, a) \sim \mathcal{N}\left(\hat{\mu}(s_t, a), \tau(a)^2/n(a)\right)$ from which then an action is again selected by maximization: $a_t = \arg\max_{a \in \mathcal{A}} \{Y(s_t, a)\}$.

In summary, in contrast to uncertainty-based exploration strategies like UCB and TS that first measure the uncertainty of the action-value estimates $\hat{\mu}(s, a)$ given the past observations, SAU-based exploration directly quantifies the uncertainty associated with each action through equation 3 by measuring the uncertainty of the sample average rewards. The clear advantage of this procedure is that it is much simpler and more computationally efficient, while at the same time providing the same quality of uncertainty estimates, and often even better exploration than the more complex algorithms that rely on estimating the uncertainty of the action-value function (see empirical results in (Rigotti & Zhu, 2021)).

## 5 $\delta^2$-EXPLORATION: SAU-BASED EXPLORATION FOR SEQUENTIAL RL

The SAU-exploration algorithms summarized in the previous section have been shown to be very effective exploration strategies despite their low computation footprint in the contextual bandit setting, including the deep contextual bandit case where the reward is being parametrized by a deep neural network (Rigotti & Zhu, 2021). We now propose an extension of SAU to adapt it from the bandit to the sequential reinforcement learning scenario. We call the resulting class of exploration algorithms $\delta^2$-*exploration*.

### 5.1 $\delta^2$-EXPLORATION IN Q-LEARNING

Contextual bandits are essentially simplified MDPs with horizon 1, so that each episode only consists of one state observation (which in contextual bandits is called "context"), an action and a reward, after which the environment is reset. Because of this relationship, the first step to extend results in bandits to sequential reinforcement learning is to substitute the bandit feedback $r_t$ in equation 3 with the multi-step cumulative return $G_t$, defined as $G_t = \sum_{k=0}^{T-t-1} r_{t+k+1}$ in the case of episodic tasks or $G_t = \sum_{k=0}^{\infty} \gamma^k r_{t+k+1}$ with discount rate $\gamma$ in the case of infinite horizon tasks. The value prediction terms, $\mu(s, a)$ in equation 3 correspond in a sequential RL setting to the state-action value $q_\pi(s, a)$ under some policy $\pi$, and can therefore be substituted with $Q(s, a)$, its current estimate at a given time.

At this point, we notice that Q-learning uses a target to approximate the cumulative return $G_t$ with the one-step iteration $r_{t+1} + \gamma \max_{a'} Q(s_{t+1}, a')$ (Sutton & Barto, 2018), and we can therefore use the same approximation in our extension of SAU from bandits to sequential RL.

Note that the SAU measure is not a function of the current context $s_t$ but just depends on action $a$. A key mechanism explaining why this simple method works is the assumption that the noise in each arm is homogeneous. This is a good approximate description of the data-generating process underlying complicated reward models such as deep neural networks, in particular when there wouldn't anyway be enough data to provide an accurate fit of uncertainty as a function of context. In sequential RL, the reward models might indeed be complicated, which bodes well for applying the SAU measure in this setting, especially in the deep RL case.

We now have everything to formulate the sequential RL version of the SAU uncertainty measure $\tau_a$ in equation 3 that we call $\delta^2$-*uncertainty*:

$$\Delta^2(a) = \frac{1}{n(a)} \sum_{t \in \mathbb{T}_a} \delta^2(s_t, a_t), \tag{4}$$

where $\delta(s_t, a_t)$ denotes the *TD-error* in equation 1 at an observed sequence $s_t, a_t, r_{t+1}, s_{t+1}$, i.e.:

$$\delta(s_t, a_t) = r_{t+1} + \gamma \max_{a' \in \mathcal{A}} Q(s_{t+1}, a') - Q(s_t, a_t),$$

and the sum in equation 4 is over the set $\mathbb{T}_a$ of time steps when action $a$ was chosen up until the current time, and $n(a)$ denotes the number of times that the action $a$ was observed.

In practice, $\Delta^2(\cdot)$ in equation 4 can be efficiently computed by incrementally updating a running estimate that accumulates the TD-error:

$$\Delta^2(a) \leftarrow \Delta^2(a) + \left[ \delta^2(s, a) - \Delta^2(a) \right] / n(a). \tag{5}$$

Comparing the derived expression equation 4 for the $\delta^2$-*uncertainty* $\Delta^2$ with the definition equation 3 of the SAU measure $\tau_a$ solidifies the intuition that the role played by the *prediction residual* $e_t$ in equation 3 for the bandit feedback is naturally taken up by the *TD-error* $\delta(s_t, a_t)$ in sequential RL.

As in the case of SAU-based exploration in bandit problems, once the $\delta^2$-uncertainty is available it can simply be plugged into the action selection process to implement a class of exploration algorithms that we call $\delta^2$-*exploration*. The $\delta^2$-exploration algorithm that is obtained by using the $\delta^2$-uncertainty as in *UCB* will be called $\boldsymbol{\delta^2}$-UCB. Analogously, the $\delta^2$-exploration algorithm obtained by combining the $\delta^2$-uncertainty with sampling-based exploration will be called $\boldsymbol{\delta^2}$-Sampling.

More specifically, given action-values $Q(s_t, a)$ at a given state $s_t$ at time $t$, $\delta^2$-exploration implements exploration by "perturbing" action-values $\widetilde{Q}(s_t, a)$ according to one of the following schemes, depending on which specific exploration algorithm we want to use:

1. $\boldsymbol{\delta^2}$-UCB: 
$$\widetilde{Q}(s_t, a) = Q(s_t, a) + \sqrt{\frac{\Delta^2(a) \log t}{n(a)}} \quad \forall a \in \mathcal{A};$$

2. $\boldsymbol{\delta^2}$-Sampling: 
$$\widetilde{Q}(s_t, a) \sim \mathcal{N}\left( Q(s_t, a), \frac{\Delta^2(a)}{n(a)} \right) \quad \forall a \in \mathcal{A}.$$

Then an action is selected by maximization: $a_t = \arg\max_{a \in \mathcal{A}} \left\{ \widetilde{Q}(s_t, a) \right\}$.

The two $\delta^2$-exploration algorithms $\boldsymbol{\delta^2}$-UCB and $\boldsymbol{\delta^2}$-Sampling are detailed in Algorithm 1.

## 5.2 COMPARISON BETWEEN VALUE UNCERTAINTY EXPLORATION AND $\boldsymbol{\delta^2}$-EXPLORATION

Let us take a moment to compare the functioning of $\delta^2$-exploration to conventional exploration strategies based on estimating uncertainty of value estimates, such as UCB and posterior sampling (Thompson Sampling) methods. We call these methods "value uncertainty exploration methods" because the mechanism that they use to promote exploration relies on quantifying the uncertainty of the current value function estimates. Maintaining this "internal uncertainty" measure is typically

---

**Algorithm 1** $\delta^2$-exploration in Q-learning.

---

**Parameters:** step size $\alpha \in (0, 1]$, discount factor $\gamma \in (0, 1]$, and $\beta \geq 1$.
**Initialize:** $Q(s, a)$ such that $Q(\text{terminal}, \cdot) = 0$, $\Delta^2(a) = 0$, and $n(a) = 1$ for all $a \in \mathcal{A}$.
**for all** episodes **do**
    Initialize state $s$
    **while** $s$ in episode is not terminal **do**
        (1) Use $\delta^2$-exploration strategy (a) or (b):

           (a) $\boldsymbol{\delta^2}$**-UCB:**
$$\widetilde{Q}(s, a) = Q(s, a) + \sqrt{\frac{\Delta^2(a) \log \sum_a n(a)}{n(a)}};$$

           (b) $\boldsymbol{\delta^2}$**-Sampling:**
$$\widetilde{Q}(s, a) \sim \mathcal{N}\left(Q(s, a), \frac{\Delta^2(a)}{n(a)}\right).$$

        (2) Select $a$ in state $s$:
$$a = \arg\max_{a' \in \mathcal{A}} \{\widetilde{Q}(s, a')\}$$
$$n(a) \leftarrow n(a) + 1.$$

        (3) Take action $a$, observe $r$, $s'$.

        (4) Update $Q(\cdot)$ and $\Delta^2(\cdot)$:
$$\delta(s, a) = r + \gamma \max_{a'} Q(s', \alpha) - Q(s, a)$$
$$Q(s, a) \leftarrow Q(s, a) + \alpha \cdot \delta(s, a)$$
$$\Delta^2(a) \leftarrow \Delta^2(a) + \left(\delta^2(s, a) - \Delta^2(a)\right)/n(a).$$

        (5) Step to next state: $s \leftarrow s'$
    **end while**
**end for**

---

computationally costly, to the point of being computationally intractable for large state spaces. Thompson Sampling in particular assumes a prior distribution over value functions that is updated based on observed transitions, a process that is only feasible for simple environments.

In contrast, rather than maintaining the internal uncertainty of value function estimates, $\delta^2$-exploration simply updates its propensity for exploration by accumulating the *reward prediction error $\delta$* as in equation 5. In other words, $\delta$ not only has a role in *policy evaluation* by representing the extent to which the value function should be update, but also has a role in the *control policy* by representing the extent to which exploration is needed to prevent the policy from being suboptimal.

Figure 1 gives a schematic overview of the difference between *value uncertainty exploration* methods and our proposed method $\delta^2$-exploration, emphasizing the simplicity of $\delta^2$-exploration whose exploration mechanism does not need to rely on estimating the distributional uncertainty over the value function, but only needs access to the reward prediction error $\delta$ to compute the $\delta^2$-uncertainty $\Delta$.

Once the $\delta^2$-uncertainty $\Delta$ is available, it is directly passed to the action selection step (see step (2) in Algorithm 1). This means that the $\delta^2$-uncertainty, besides being cheap to compute, is also simple to use to instantiate the exploration mechanism: it is simply used to "perturbe" the estimated action-values (either as an exploration bonus in the case of $\delta^2$-UCB or as sampling noise in the case of $\delta^2$-Sampling) before greedily selecting the next action. This underscores that from a computational standpoint the action selection process of $\delta^2$-exploration is as simple as $\epsilon$-greedy action selection (equation 2). Indeed, $\delta^2$-exploration inherits this trait from SAU-exploration (Rigotti & Zhu, 2021), and similarly to that algorithm can be used as a flexible *drop-in replacement* for $\epsilon$-greedy selection.

## 5.3   $\boldsymbol{\delta^2}$-EXPLORATION IN DEEP Q-LEARNING

Deep Q-learning consists in parameterizing the action-value function $Q(s, a)$ as a deep neural network, i.e. a Deep Q Network (DQN) *policy network* (Mnih et al., 2015). Because of the complexity of such an action-value function, it is notoriously difficult to implement efficient exploration strategies in DQN. As a result, dithering heuristics such as $\epsilon$-greedy are commonly used in practice. Fortunately,

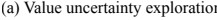

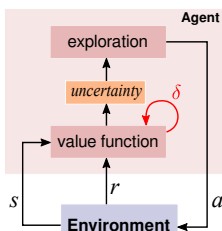

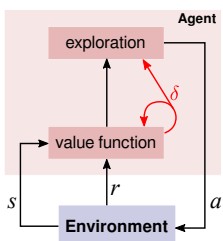

Figure 1: (a) *Value uncertainty exploration algorithms* like UCB and Thompson Sampling rely on an estimate of the *uncertainty* of the current *value function*, a quantity that is intractable already for moderately complex value functions and environments. (b) $\delta^2$-exploration on the other hand only needs access to the *reward prediction error* $\delta$ to compute how to select an action that trades off exploration and exploitation in a principled way.

$\delta^2$-exploration just depends on the prediction of the value function, so it is versatile enough to be easily applicable to DQN.

A direct way to combine $\delta^2$-exploration with DQN would be to implement Algorithm 1 using a deep neural network (in addition to the policy network) that would fit the $\delta^2$-exploration measure $\Delta$. To avoid the computational and memory overhead of an additional network we took inspiration from recent papers that harnessed the *target network* in DQN (a delayed parameter version of the policy net used in Mnih et al. (2015) to stabilize training) in order to implement unbiased estimators of the value (van Hasselt et al., 2016; Zhu & Rigotti, 2020). We reasoned that the difference between the online policy net $Q(s, a)$ and the delayed parameter target net $Q^-(s, a)$ should reflect a minimization of the reward prediction error. This suggests to replace the expression equation 4 with the following expression for the $\delta^2$-uncertainty $\Delta$:

$$\Delta_{\text{dqn}}^2(s, a) = \left( \max_{a'} Q(s, a') - Q^-(s, a) \right)^2.$$

We propose to use this variation of the $\delta^2$-uncertainty to implement the DQN version of $\delta^2$-exploration algorithms **$\delta^2$-UCB** and **$\delta^2$-Sampling**. Using $\delta^2$-exploration in DQN therefore comes down to implementing DQN learning as usual (see e.g. Mnih et al. (2015)), except that the action selection part of the algorithm (which typically would use $\epsilon$-greedy action selection) is replaced as in Algorithm 2.

---

**Algorithm 2** $\delta^2$-exploration in Deep Q-Networks.

---

1: **Initialize:** Regular DQN (Mnih et al., 2015) with *policy network* $Q(s, a)$, *target network* $Q^-(s, a)$, and global action counts $n(a)$ initialized at 1 for all $a \in \mathcal{A}$.
2: **Replace** action selection ACT($\cdot$) in regular DQN with:

3: **function** ACT-$\delta^2$-EXPLORATION($s$)
4:     **for** $a = 1, \ldots, |\mathcal{A}|$ **do**

5:         (1) Compute       $\Delta_{\text{dqn}}^2(s, a) = \left( \max_{a'} Q(s, a') - Q^-(s, a) \right)^2$

6:         (2) Use $\delta^2$-exploration strategy (a) or (b):

7:             (a) $\delta^2$-UCB:     $\widetilde{Q}(s, a) = Q(s, a) + \sqrt{\dfrac{\Delta_{\text{dqn}}^2(s, a) \log \sum_a n(a)}{n(a)}};$

8:             (b) $\delta^2$-Sampling:     $\widetilde{Q}(s, a) \sim \mathcal{N}\left( Q(s, a), \dfrac{\Delta_{\text{dqn}}^2(s, a)}{n(a)} \right).$

9:     **end for**
10:     Select action $a^* = \arg\max_{a \in \mathcal{A}}\{\widetilde{Q}(s, a)\}$.
11:     Update global count $n(a^*) \leftarrow n(a^*) + 1$.
12:     **Return** action $a^*$
13: **end function**

---

## 6 EMPIRICAL RESULTS

We now empirically evaluate the performance of our proposed $\delta^2$-exploration strategies in a series of reinforcement learning tasks, including classical tabular Q-learning and Deep Q Learning tasks.

We will benchmark $\delta^2$-exploration against $\epsilon$-greedy action selection, the *de facto* go-to exploration strategy in Q-learning, and also the closest exploration strategy in terms of computational and implementation efficiency. In fact, we propose $\delta^2$-exploration as a flexible drop-in replacement for $\epsilon$-greedy action selection that we now demonstrate is remarkably more efficient, despite being comparably simple in terms of computation and implementation complexity.

### 6.1 Q-LEARNING ON CLIFF-WALKING TASK

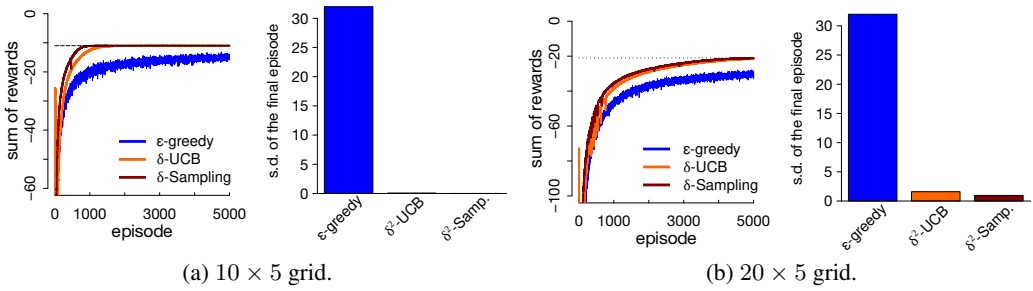

(a) $10 \times 5$ grid.  (b) $20 \times 5$ grid.

Figure 2: Cliff-walking RL task. Both $\delta^2$-exploration algorithms achieve higher cumulative reward at a faster rate than $\epsilon$-greedy action selection for both grid sizes, even attaining the optimal reward (indicated by the black dotted line). Moreover, as the bar plots show, $\delta^2$-exploration reaches a lower variability of received rewards, which indicates that it is more stable. $\epsilon$-greedy exploration with $\epsilon = 1/\sqrt{n(s)}$, where $n(s)$ is the number of times state $s$ has been visited. Learning rate is set to $\alpha_n(s, a) = 0.1(100 + 1)/(100 + n(s, a))$, where $n(s, a)$ is the number of updates of each state-action. Data points are averaged over 1000 runs.

We evaluate the performance of $\delta^2$-exploration on the cliff-walking task (Example 6.6 in Sutton & Barto (2018)), a standard undiscounted episodic task with start and goal states, and four movement actions: up, down, right, and left. Reward is $-1$ on all transitions except those into the "Cliff" region (bottom row except for the start and goal states). If the agent steps into this region, she gets a reward of $-100$ and is instantly sent back to the start. We vary the environment size by considering $10 \times 5$ and $20 \times 5$ grids.

We measure performance as cumulative reward during episodes, and report average values for 1000 runs and standard deviation of values of the final episode in Fig. 2. We compare our two proposed $\delta^2$-exploration $\delta^2$-UCB and $\delta^2$-Sampling against $\epsilon$-greedy action selection with $\epsilon = 1/\sqrt{n(s)}$. As Fig. 2 shows the $\delta^2$-exploration strategies perform much better than $\epsilon$-greedy, quite quickly attaining near optimal values (the reard value indicated by the black dotted lines in the plot). We also compute the standard deviation of rewards of the final rewards of 1000 runs, which show that $\delta^2$-exploration strategies display much lower variability (i.e. are more stable) than $\epsilon$-greedy action selection.

### 6.2 DEEP Q NETWORK ON THE ATARI 2600 DOMAIN

As a final empirical validations for $\delta^2$-exploration we study the performance of our DQN adaptation Algorithm 2 on 8 representative games from the Atari 2600 domain (Bellemare et al., 2013). We choose 4 "easy" games for which DQN displays super-human performance, and 4 "hard" games for which DQN performs around or below humans. We compared the performance of the $\delta^2$-exploration algorithms with the conventional $\epsilon$-greedy action selection that consists of annealing $\epsilon$ for the first 1M steps from 1.0 linearly to $\epsilon_F = 0.1$, noisy nets (Fortunato et al., 2017) and Bootstrapped DQN (Osband et al., 2016), a strong baseline, but which requires a form of ensembling resulting in considerable memory overhead. Fig. 3 compares the performance at the end of training for $\epsilon$-greedy, Bootstrapped DQN, noisy nets and our $\delta^2$-exploration algorithms. The advantage of $\delta^2$-exploration is particularly remarkable for harder games, and in particular *MsPacman*, where our algorithms finds

policies that reach scores that are almost double what the other competitors reach, and *Centipede*, where $\delta^2$-Sampling vastly outperforms even Bootstrapped DQN.

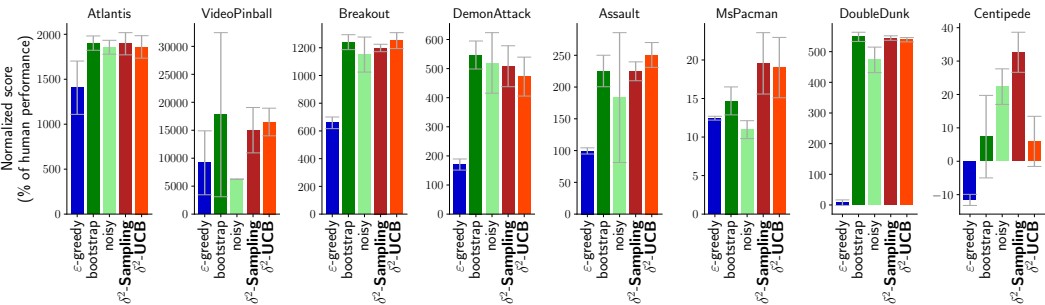

Figure 3: $\delta^2$-exploration DQN policies consistently outperform $\epsilon$-greedy exploration and noisy nets and are on par and sometimes better than the strong but computationally costly Bootstrapped DQN in the Atari 2600 domain. DQN training is done for 20M steps for all games, except VideoPinball which is trained for 80M steps (see Appendix A). Performance is then computed as the average score over the last 1M frames of training. Bars indicate score normalized to human performance (see Mnih et al. (2015)) across 6 random seeds, error bars are SEM. $\delta^2$-UCB and $\delta^2$-Sampling exploration are compared to $\epsilon$-greedy exploration with $\epsilon$ linearly decreasing over 1M steps starting from 1.0 to $\epsilon_F = 0.1$, as standard (Mnih et al., 2015). Both $\delta^2$-exploration strategies reach consistently competitive policies at the end of training.

## 7 CONCLUSIONS AND DISCUSSION

Here we extended Sample Average Uncertainty, a recently proposed uncertainty measure for exploration, from bandit problems to the general sequential reinforcement learning setting. This resulted in our $\delta^2$-exploration algorithms, two exploration strategies that are extremely simple and flexible to implement, as they only depend on the predicted action-values. These can be used as drop-in replacements for $\epsilon$-greedy action selection in Q-learning and DQN, resulting in effective and computationally efficient learning and superior empirical performance over the $\epsilon$-greedy baseline.

Our work provides a simple, scalable and robust exploration algorithms, that can be effortlessly deployed to mitigate the exploration-exploitation dilemma. While the resulting consequences of the deployment of an RL algorithm, ethical or otherwise, will clearly depend on the specific applications, we believe that providing efficient uncertainty estimates and more reliable exploration than the common $\epsilon$-greedy action selection strategy will potentially have a positive contribution in terms of sample-efficiency and optimality of the learned policies, as well as in terms of their trustworthiness.

We limited our experiments to comparisons against $\epsilon$-greedy exploration, given its importance and ubiquity, and because our method was developed specifically as a natural drop-in replacement for $\epsilon$-greedy exploration. Our $\delta^2$-exploration algorithms have in fact a comparably low computation footprint and can be implemented through a minimal API that is the same as for $\epsilon$-greedy exploration, only needing to access the action selection process of the RL agent. This enables modular implementations with minimal change on existing RL code bases, as for instances exemplified by the DQN version of $\delta^2$-exploration in Algorithm 2, which simply consists in the replacement of the action selection method in a conventional DQN with no modification of the training algorithm. This is in stark contrast with previous exploration strategies based on quantifying value uncertainty, which typically impose limitations on the representation of the value function, need to heavily modify the evaluation process, and introduce costly training loops to update the value uncertainty.

That said, the remarkable performance of our simulation results and the flexibility of $\delta^2$-exploration suggest that it could be applied to even more sophisticated RL settings and possibly combined with other learning algorithms besides Q-learning and DQN, an option that we are looking forward to investigating in the future.

REPRODUCIBILITY STATEMENT

We provide detailed pseudo-code of our proposed algorithms and tried our best to indicate important hyperparameters to replicate our experiments and reproduce our benchmarks, which are anyway based on tasks that are already publicly available. In addition, upon acceptance of our paper, we plan to release the code to replicate all our numerical experiments on github.

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

## A    ADDITIONAL FIGURE

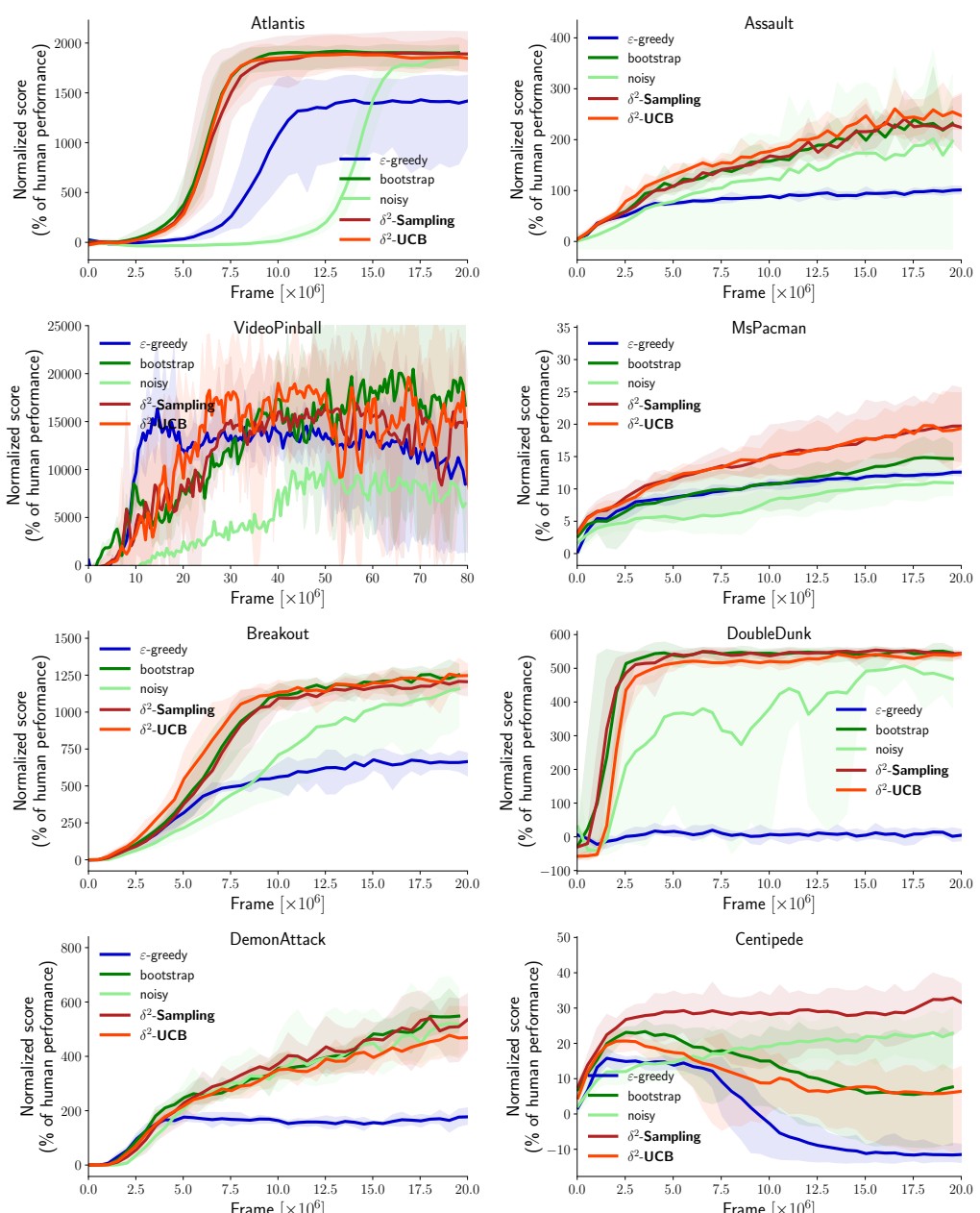

Figure 4: Learning curves of $\epsilon$-greedy exploration, bootstrapDQN, noisy nets, $\delta^2$-UCB and $\delta^2$-Sampling on 4 "easy" and 4 "hard" (i.e., performance of DQN is respectively higher and lower than that of human players) additional Atari 2600 games. Scores are averaged in intervals of 500k frames, plots indicate mean score over 6 independent random seeds, and shaded areas indicate min and max scores over the 6 seeds. $\delta^2$-exploration algorithms consistently have scores among the highest throughout learning compared to the $\epsilon$-greedy strategy, noisy nets and bootstrapDQN. Remarkably, for two "hard games", *Assault* and *DoubleDunk*, for which $\epsilon$-greedy only reaches sub-human performance, $\delta^2$-exploration can very rapidly find policies that beat human players by more than 200% and 500%, respectively, rivaling the performance of bootstrapDQN. For *VideoPinball* and *Centipede* where $\epsilon$-greedy seems unstable, $\delta^2$-exploration strategies are able to maintain better stability throughout learning, even beating bootstrpDQN in some cases like *Centipede*.

