# OpenReview forum: "$\sbf{\delta^2}$-exploration for Reinforcement Learning"
_ICLR.cc/2022/Conference — ICLR 2022 Submitted_

### Official Review · Reviewer_uzFB · 2021-10-28

**Correctness:** 2
**Technical Novelty And Significance:** 2
**Empirical Novelty And Significance:** 1
**Recommendation:** 3
**Confidence:** 4

**Main Review:**

The method is simple and intuitive, and I think it's an interesting idea. However, I have various concerns about the paper.

The paper does not try to claim any connection with theory, but I'm wondering what is the intuition behind this approach. In particular, the idea of averaging over states seems very suboptimal in RL. In particular, it is easy to imagine problems where you have only a few actions available (in the extreme case one) in certain states that are visited often. By using your method you are decreasing the uncertainty of these actions (by dividing by the total counter $n(a)$) also in states that are maybe hard to reach, potentially leading to under-exploration. I understand that the numerator is a squared error, but it still seems potentially critical. It would be good to test your method in other tabular domains where this situation may happen.

This leads me to the second issue: experiments are not enough to support your claims. First, I disagree with the argument that you need only to compare with the $\epsilon$-greedy strategy. Second, there is not enough diversity in terms of domains and baselines to support your claims.
There are plenty of easy methods that could be tested. First of all, you could have tested multiple annealing schedules for $\epsilon$-greedy and not only $1/\sqrt{n(s)}$, e.g., $n(s)^{\alpha}$ with $\alpha>0.5$ or even action-dependent values $n(s,a)^{\alpha}$. There are also other techniques, like optimistic initialization of the Q-function, that often help with exploration. I would have also tested softmax exploration, this is another standard technique for randomized exploration with Q-learning. There are also inverse gap techniques, originally introduced in [1]. This is just a short list of easy techniques you could compare with. Clearly, in tabular RL, you could even try to compare with exploration strategies based on upper-confidence bounds or Thompson sampling.

I found also limited the set of experiments in deepRL. Several heuristics have been proposed over time and I believe it is necessary to compare with a few of them even if they are more complicated. It is important for understanding the potential sub-optimality of the proposed strategy.

Moreover, I would add more ablation studies either in tabular and continuos MDPs. For example, you can play with the level of stochasticity in the MDP, the structure of the MDP itself (as mentioned in a previous point) and add other classical control environments.

Finally, you refer to the appendix that seems not to be available. Maybe we could have obtained more details about the setting used in the experiment (or other experiments) from there.

[1] Abe and Long, Associative reinforcement learning using linear probabilistic concepts. ICML 1999

Minor comments:
- I would suggest adding a dependence on the current time $t$ in the definition of $\Delta^2(a)$ and providing an explicit definition of $\mathbb{T}_a$

**Summary Of The Paper:**

The paper extends the recently introduced SAU measure from bandits to RL. The idea is to enhance the exploration of action based on an approximation of the estimation error. More precisely, the bonus is proportional to the average squared temporal difference error. This uncertainty measure is easy to compute and can be integrated into both tabular and continuous methods (e.g., Q-learning and DQN).

**Summary Of The Review:**

Overall, while I think the idea is interesting, it is necessary to provide a more detailed experimental evaluation of the method to support the claims. I think the paper is not currently ready for publication.

---

> ### Author Response · Authors · 2021-11-23
> **Response to Reviewer uzFB**
>
> We thank the Reviewer for taking the time of reading our work and providing detailed comments. Here below we address the main points raised by the Reviewer.
>
> About the idea of averaging over states: this comes from the SAU algorithm, an uncertainty quantification method very recently developed in the bandit literature. As it turns out, we also considered a version of the algorithm where Delta=Delta(s,a), i.e. it depends on state-action pair, which performs very similarly to the Delta(a) version. In the revisions of the paper we will supply these additional experiments and clarify differences between the two version, and speculate on why the Delta(a) version might be working as well as the Delta(s,a) version of the algorithm. In short, our view is that an initial exploration phase can be though of as a low data regime, where there is not enough data to take full advantage of a state-dependent uncertainty model and a lower capacity uncertainty model that only depends on the action might work as well. In the revisions we aim to supply the comparisons between the version of delta^2 dependent only on action or state-action pairs, and clarify all the points that were just discussed.
>
> About the specific annealing schedule for epsilon-greedy: we considered other choice of annealing schedule, such as $n(s)^\alpha$, but in each case our method obviously outperformed over epsilon-greedy. The main aim of our study was to focus on exploration, and develop a simple and fairly generic algorithm that could be used as a drop-in replacement for epsilon-greedy exploration. For this, we thought that it made sense to, at least initially, benchmark delta^2-exploration mainly against epsilon-greedy exploration since they have the same level of computational complexity and memory requirements, as opposed to much more computationally involved methods such a TS.

---

> > ### Comment · Reviewer_uzFB · 2021-11-23
> > **Follow up**
> >
> > Thank you for the answers. While I think it is an interesting idea, I still believe that the paper needs a "major" revision to incorporate reviewers' concerns and extend the empirical analysis. As mentioned in my review, I think there are many different strategies that are as simple as $\epsilon$-greedy that can be evaluated to make the experiments more robust. This is quite important since the paper lacks theoretical support.
> >
> > Is the appendix missing in the original submission?

---

> > > ### Author Response · Authors · 2021-11-23
> > > **The appendix is on Page 12 (the last page).**
> > >
> > > Sorry, we missed this question.
> > > The appendix is on Page 12 (the last page).

---

### Official Review · Reviewer_PFBt · 2021-10-29

**Correctness:** 3
**Technical Novelty And Significance:** 1
**Empirical Novelty And Significance:** 2
**Recommendation:** 3
**Confidence:** 4

**Main Review:**

The paper are well-written and easy-to-understand. However, there are following concerns from my point of view.

The algorithm looks like a trivial extension of SAU method developed by Rigotti & Zhu. There should be more discussion about the difference between $\delta^2$-exploration method and SAU method. To me, it seems the technical novelty of the algorithm design is rather weak.

The authors fail to explain why this simple exploration strategy can work (or at least better than $\epsilon$-greedy). As far as I am concerned, collecting enough samples for an action on certain states cannot indicate well-exploration of such an action on any other states. There are many simple hard instances that this strategy may fail. For example, we can construct an MDP in which the agent will always enters state $s_1$ with high probability, and occasionally enters $s_2$. The expected reward of $a_1$ is higher than $a_2$ in $s_1$, while the expected reward of $a_2$ is higher than $a_1$ in $s_2$. Using $\delta^2$-exploration strategy, the agent may never take action $a_2$ in $s_2$ if the expected reward of $(s_2,a_1)$ is higher than $(s_2,a_2)$ in the first several attempts. Therefore, I believe the exploration strategy is not theoretically-convinced enough to obtain good performance on all possible RL tasks.

The experiments in the paper seems not enough to indicate the benefits of $\delta^2$ exploration mechanism. The authors evaluate their algorithm on the Cliff-Walking task and the Atari games. For the former, the reason why the $\epsilon$-greedy algorithm achieves slightly worse performance may be that the agent may occasionally choose the bad action due to the $\epsilon$-greedy mechanism. This case may not always happen in the general RL tasks. For the latter, the authors only conduct experiments on 8 games. I believe that more experiments are necessary (on other games or even other benchmarks such as Mujoco tasks) if the authors want to argue that $\delta^2$-exploration could serve as a competitive baseline or even replace the role of $\epsilon$-greedy in RL. Also, I doubt whether these games are particularly selected or randomly chosen.

**Summary Of The Paper:**

This paper extends a recently-proposed exploration method called SAU in bandits to the RL problem. They combine this exploration approach with the standard Q-learning algorithm. Their experiments show that this approach can obtain better performance compared with the Q-learning algorithm with eps-greedy exploration.

**Summary Of The Review:**

Based on the above issues, I am inclined to given a negative score. I will increase the score if the above problems are well-tackled in the rebuttal:
1. Discussions on the comparison with SAU method by Rigotti & Zhu.
2. Explanation of the insight behind the exploration strategy.
3. More experiments to support their conclusion.

---

> ### Author Response · Authors · 2021-11-23
> **Response to Reviewer PFEt**
>
> We thank the Reviewer for taking the time of reading our paper and for providing concrete point-by-point suggestions for improving our paper. Here below we address individually these suggestions.
>
>  1. We will be happy to discuss more in depth the connections between our delta^2-exploration algorithm and it’s original inspiration, the SAU algorithm developed in the bandit setting. Investigating bandits and RL is byitself a worthy endeavor, because bridging the application and methodological gap between the two fields could result in useful insights and algorithms in both. Many exploration methods have been developed in for bandits, but directly applying them to RL is often impractical. For example, there is not practical way to directly apply TS to RL.
> One of the main difficulties in bridging the gap between bandits and RL is that, due to the transition probabilities from the state to next state, reward generation process is much more complicated than what is assumed in the bandit setting. However, despite this crucial difference, our intuition is that that RL is fundamentally about solving the prediction problem. In this sense, it is similar to the bandit framework. Thus, when an agent takes an action, the agent needs to consider the uncertainty due to the prediction error. This intuition motivates the application of SAU-based exploration in RL.
>
>  2. As the Reviewer points out (and also others pointed out in their reviews) the version of delta^2-exploration that we presented has a measure of uncertainty that solely depends on the action. This comes from the theoretical roots of this works that originates in the SAU-exploration algorithm that was developed to tackle the bandit setting. As it turns out, we also considered a version of the algorithm where Delta=Delta(s,a), i.e. it depends on state-action pair, which performs very similarly to the Delta(a) version. In the revisions of the paper we will supply these additional experiments and clarify differences between the two version, and speculate on why the Delta(a) version might be working as well as the Delta(s,a) version of the algorithm. In short, our view is that an initial exploration phase can be though of as a low data regime, where there is not enough data to take full advantage of a state-dependent uncertainty model and a lower capacity uncertainty model that only depends on the action might work as well. This is related to the homogenous error assumption that the SAU paper used to motivate the same use of an uncertainty measure that does not depend on state (context) also in the Contextual Bandit case.
>
> In our delta^2-exploration, the key part is in the SAU measure (defined by Delta^2/n) which includes two parts: (1) the denominator counts how many times an action (or state-action pair) has been played. This part is related to count-based exploration. (2) the numerator represents the uncertainty heterogeneity among actions (or state-action pairs).
> In the revisions we aim to supply the comparisons between the version of delta^2 dependent only on action or state-action pairs, and clarify all the points above.
>
> 3. The Cliff-walking task is a good illustrative task for showing why our method is better than epsilon-greedy, since epsilon-greedy occasionally chooses the bad action with catastrophic consequences. Occasional selection of the suboptimal action is something that has consequences that are particularly clear in the Cliff-walking tasks, but it important to point out that this is actually exemplary of a general shortcoming of epsilon-greedy, since it uniformly consider all suboptimal choice.
>
> About the choice of the Atari games, they are not chosen according to any particularly order, other than wanting to tackle a good range of difficulties (quantified in terms of the relative performance of DQN over human players). In particular, we are showing results for all the games we tried, i.e. we didn't leave out games that had been tried but showed lower performance than desirable. The main reason for not systematically testing all games of the Atari 2600 suite is the considerable computation cost of doing so, considering that each game has to be played with all the baseline algorithms, and in addition has to be played several times with multiple seeds to assess repeatability. In any case, in the revisions we plan to supply results on a few more games.

---

### Official Review · Reviewer_o8hT · 2021-10-30

**Correctness:** 3
**Technical Novelty And Significance:** 3
**Empirical Novelty And Significance:** 2
**Recommendation:** 5
**Confidence:** 3

**Main Review:**

Strength:

1. This work proposes an exploration strategy that does not hinge on uncertainty estimates, which are typically computationally expensive and require either ensemble of models or specifically designed neural nets.

2. This work incorporates recent advances in the theoretical study of bandits, which are promising and have been shown to work effectively by various previous studies. e.g. IDS, UCB, Posterior Sampling

Weakness:

1. The work lacks theoretical support. Though incorporating SAU for exploration is promising (as I claimed in the strength part), the proposed incorporation method in this paper is not grounded by theory. Particularly, in Algorithm 1 of this paper, the author directly incorporates SAU for exploration in Q learning, where the SAU hinges only on the actions. It makes sense to estimate uncertainty based only on the novelty of actions under the multi-arm bandit setting. Nevertheless, in the contextual bandit setting, SAU requires a notion of homogeneous context to yield a reasonable uncertainty estimate [1]. Thus, it is questionable if directly incorporating SAU for the more complicated RL setting is provably efficient.

2. In Algorithm 2, it is reasonable to use deviation between the target and the Q network to compute the TD error. However, it seems that $\Delta_{dqn}$ no longer aggregates the TD errors according to the actions taken as SAU does. Alternatively,$\Delta_{dqn}$ is defined for each of the individual state-action pairs. It would be better if the author could give some discussion on such an adjustment in DQN implementation.

In addition, some exploration strategies based on Q-learning are missing in the literature review. For instance, exploration with IDS ([2], [3]) is also an important SOTA, which incorporates bandit exploration strategy into RL exploration. In addition, from the empirical perspective, previous works also incorporate curiosity-driven methods [4] and mutual information ([5],[6],[7]) to construct the exploration bonus.

[1]  Zhu and Rigotti, Deep Bandits Show-Off: Simple and Efficient Exploration with Deep Networks. (2021)

[2] Russo and van Roy, Learning to Optimize Via Information-Directed Sampling. (2017)

[3] Nikolov et al., INFORMATION-DIRECTED EXPLORATION FOR DEEP REINFORCEMENT LEARNING. (2019)

[4] Pathak et al., Curiosity-driven Exploration by Self-supervised Prediction. (2017)

[5] Houthooft et al., VIME: Variational Information Maximizing Exploration. (2017)

[6] Kim et al., EMI: Exploration with Mutual Information. (2017)

[7] Kim et al., Curiosity-Bottleneck: Exploration by Distilling Task-Specific Novelty. (2019)




**Summary Of The Paper:**

This work introduces $\delta^2$-exploration for reinforcement learning (RL), which aims to incorporate sample average uncertainty (SAU) into RL exploration. The authors discussed the background and formulation of SAU. The authors further propose $\delta^2$-exploration, which incorporates SAU into Q-learning and compare such exploration with value-uncertainty exploration (UCB-type exploration) and $\epsilon$-greedy. The author then proposes to incorporate $\delta^2$-exploration into DQN and conduct experiments to compare $\delta^2$-exploration with SOTA exploration algorithms. Empirical results show that $\delta^2$-explorations attain comparable results to bootstrapped DQN.

**Summary Of The Review:**

The idea of incorporating SAU into RL exploration is promising. Nevertheless, the presence of context and transition dynamics makes it unclear from the theoretical perspective whether SAU can be directly adopted into the exploration of RL. Hence, the work would be stronger if the authors could provide additional quantitative arguments supporting the validity of SAU for RL. In addition, the comparison with other baselines can be made more straightforward if the authors could provide charts of evaluation scores.

---

> ### Author Response · Authors · 2021-11-23
> **Response to Reviewer o8hT**
>
> We want to thank the Reviewer for providing in-depth comments on our work. Here below we are addressing one by one the weaknesses pointed out by the Reviewer, we hope in a satisfactory way.
>
> 1. As the Reviewer points out the version of delta^2-exploration that we presented has a measure of uncertainty that solely depends on the action. This comes from the theoretical roots of this works that originates in the SAU-exploration algorithm that was developed to tackle the bandit setting. As it turns out, we also considered a version of the algorithm where Delta=Delta(s,a), i.e. it depends on state-action pair, which performs very similarly to the Delta(a) version. In the revisions of the paper we will supply these additional experiments and clarify differences between the two version, and speculate on why the Delta(a) version might be working as well as the Delta(s,a) version of the algorithm. In short, our view is that an initial exploration phase can be though of as a low data regime, where there is not enough data to take full advantage of a state-dependent uncertainty model and a lower capacity uncertainty model that only depends on the action might work as well. This is related to the homogenous error assumption that the SAU paper used to motivate the same use of an uncertainty measure that does not depend on state (context) also in the Contextual Bandit case.
>
> 2. We thank the Reviewer for pointing out this occasion to clarify our work. Because the complexity of the environments in the Atari 2600 benchmark, and in particular the strong dependence of action outcome on a high-dimensional state, the homogenous error assumption used in the SAU paper may be too strong. Thus, in this case we purposefully developed an exploration model that depends on state-action pair. We will be happy to clarify these points in the revisions, along with further considerations on the effect of aggregating TD errors.
>
> We thank the Reviewer for the list of related papers. We will be happy to cite them and discuss them in relation to our work in the revised version of the paper.

---

### Official Review · Reviewer_zyS1 · 2021-11-01

**Correctness:** 2
**Technical Novelty And Significance:** 2
**Empirical Novelty And Significance:** 3
**Recommendation:** 5
**Confidence:** 4

**Main Review:**


#### Strengths
1. SAU has good theoretical justification in bandits. This paper contributes to connecting the RL theory and empirical RL, two largely separated areas. I think this should be encouraged.
2. The algorithm is simple and can potentially be extended to other methods such as policy gradient.

#### Weakness
1. The experiments are mainly compared against vanilla DQN. Since the authors' algorithm is a modified DQN, it would be fairer to compare their algorithm against other modified DQN variants, such as Rainbow, etc. In particular, the authors should compare with DQN with random network distillation (RND), because RND can also be interpreted as adding a UCB-type bonus and thus share similarities with the authors' algorithm.
2. The experimental results seem not improve much upon existing algorithms in RL.
3. It is strange to use the same counter $n(a)$ across different states, instead of a state-dependent one, e.g. $n(s, a)$. This does not make much sense. May the authors further justify this part? Or provide further ablation studies, e.g. fixing $\delta^2 \equiv 1$, to see if the performance of their algorithm is indeed due to the SAU?

**Summary Of The Paper:**

This paper studies exploration bonus in practical deep RL based on Sample Average Uncertainty (SAU) and upper confidence bound (UCB). SAU is a recently studied novel uncertainty quantification that works for rather arbitrary estimators. Previous paper has studied how to use SAU to derive UCB-type bonus in multi-armed bandits and proved that it could achieve optimal regret.

This paper successfully extends the SAU-UCB-type exploration bonus from bandits settings to RL and most importantly, deep RL settings, and shows how to incorporate SAU-UCB-type bonus in (deep) RL. This paper conducts various experiments for RL and deep RL, demonstrating the advantage of their algorithms over the standard benchmarks.

The paper is generally well-written.

**Summary Of The Review:**

I think the novelty outweighs the weakness, and I would recommend weak accept.

**After Discussion:** I read other reviewers' reviews. Weakness 1 pointed out by Reviewer o8hT mentioned a paper that seems to contradict this paper [Zhu and Rigotti, 2021], and the authors seem not responding to this concern. Also, Reviewer uzFB pointed out several other baselines, including Boltzmann exploration and inverse-gap, that the authors did not compare with. I think the first concern downgrades my evaluation on the theoretical strength of this paper, and the second concern downgrades the empirical strength. Thus, I would like to recommend weak reject.

---

> ### Author Response · Authors · 2021-11-23
> **Response to Reviewer zyS1**
>
> We would like to thank the Reviewer for taking the time to review our work and providing encouraging comments.
> Here below we individually address the weaknesses pointed out in the review:
>
> 1. The main aim of our study was to focus on exploration, and develop a simple and fairly generic algorithm that could be used as a drop-in replacement for epsilon-greedy exploration. In particular, our algorithm can be combined with any modification of DQN and replace their exploration method.
> For this, we thought it made sense to, at least initially, benchmark delta^2-exploration in conjunction with a vanilla DQN model, to quantify the increase in performance due to the change in the exploration module, all other things being kept equal.
> As for the comparison with RND, it is indeed a method that is related to ours. However, we did not consider a direct comparison because its original version does not provide a combination of RND and Q-learning (or DQN). We will consider this comparison when we extended delta^2-exploration to other methods, such as AC algorithms.
>
> 2. As the reviewer notes, our method does not improve much over Boostrapped DQN. However, our method is conceived as a low-cost alternative for exploration in Q-learning or DQN, which in particular doesn’t need to keep track of a whole ensemble of models like Bootstrapped DQN, and indeed has a computational and memory footprint and ease of implementation that is comparable to epsilon-greedy greedy.
>
> 3. In the bandit framework, SAU is assumed to be just dependent on actions (independent of states) due to homogenous error assumption made in that work. We initially adopted the same premise in the implementation on Q-learning. But we also considered a version of delta^2-exploration that, as suggested by the Reviewer, depends on state-action pair. As it turns out, both versions work equally well. In the revisions we will supply both version and provide our speculations as to why even a version without state dependence could work well in the low data regime (where there is not enough data to fit a state-dependent uncertainty model) such as during initial exploration.
> Moreover, in DQN using the version of the algorithm with n(a) instead of n(s,a) has the additional advantage of not needing to have to model n(s,a),  which for high-dimensional state spaces like in Atari 2600 is clearly problematic.
>
> With regards to the suggested experiment of setting delta=1 to see if the performance of our algorithm is indeed due to SAU, we performed that experiment. The results support our take that the performance improvement is indeed due to SAU. We want to thank the Reviewer for suggesting this experiment and we plan to supply its results in the revisions.

---

> > ### Comment · Reviewer_zyS1 · 2021-11-26
> > **Thanks**
> >
> > Thanks for the reply!

---

### Decision · Program_Chairs · 2022-01-20

**Decision:**

Reject

**Comment:**

Although the reviewers found the idea of the work interesting, they all think it is not ready for publication. The experiments do not properly support the claims. Discussion on the connection to some related work is missing. And also the proposed method is not well motivated. I suggest the authors to take the reviewers' comments into account, revise their work and prepare it for future venues.